# Targeting Mitochondrial Calcium Uptake with the Natural Flavonol Kaempferol, to Promote Metabolism/Secretion Coupling in Pancreatic β-cells

**DOI:** 10.3390/nu12020538

**Published:** 2020-02-19

**Authors:** Flavien Bermont, Aurelie Hermant, Romy Benninga, Christian Chabert, Guillaume Jacot, Jaime Santo-Domingo, Marine R-C Kraus, Jerome N. Feige, Umberto De Marchi

**Affiliations:** Nestlé Research—EPFL Innovation Park, CH-1015 Lausanne, Switzerland; colorsonfb@gmail.com (F.B.); aurelie.hermant@rd.nestle.com (A.H.); romybenninga@gmail.com (R.B.); christian.chabert@rd.nestle.com (C.C.); guillaumeeric.jacot@rd.nestle.com (G.J.); jaime.santodomingo@rd.nestle.com (J.S.-D.); marine.kraus@rd.nestle.com (M.R.-C.K.); jerome.feige@rd.nestle.com (J.N.F.)

**Keywords:** mitochondria, calcium, insulin, polyphenols, β-cell, kaempferol, mitoxantrone

## Abstract

Pancreatic β-cells secrete insulin to lower blood glucose, following a meal. Maintenance of β-cell function is essential to preventing type 2 diabetes. In pancreatic β-cells, mitochondrial matrix calcium is an activating signal for insulin secretion. Recently, the molecular identity of the mitochondrial calcium uniporter (MCU), the transporter that mediates mitochondrial calcium uptake, was revealed. Its role in pancreatic β-cell signal transduction modulation was clarified, opening new perspectives for intervention. Here, we investigated the effects of a mitochondrial Ca^2+^-targeted nutritional intervention strategy on metabolism/secretion coupling, in a model of pancreatic insulin-secreting cells (INS-1E). Acute treatment of INS-1E cells with the natural plant flavonoid and MCU activator kaempferol, at a low micromolar range, increased mitochondrial calcium rise during glucose stimulation, without affecting the expression level of the MCU and with no cytotoxicity. Enhanced mitochondrial calcium rises potentiated glucose-induced insulin secretion. Conversely, the MCU inhibitor mitoxantrone inhibited mitochondrial Ca^2+^ uptake and prevented both glucose-induced insulin secretion and kaempferol-potentiated effects. The kaempferol-dependent potentiation of insulin secretion was finally validated in a model of a standardized pancreatic human islet. We conclude that the plant product kaempferol activates metabolism/secretion coupling in insulin-secreting cells by modulating mitochondrial calcium uptake.

## 1. Introduction

Targeting pancreatic β-cells is a promising strategy to treat diabetes, due to the crucial role of the pancreatic β-cells in the pathogenesis of both type 1 and type 2 diabetes. Therefore, preservation, expansion, or improved function of β-cells are current approaches for targeting this cell type in the management of diabetes. The modulation of the biological pathways that regulate β-cell function represents the next stage of discovery in this field [1]. 

Within this framework, targeting mitochondrial Ca^2+^ in pancreatic β-cells is emerging as an innovative and promising strategy for diabetic patients. Indeed, mitochondrial matrix Ca^2+^ has been demonstrated to be an activating signal for insulin secretion [2]. The proposed mechanism of mitochondrial Ca^2+^-dependent pancreatic β-cell stimulation is based on the coordinated activation of two mitochondrial processes—oxidative metabolism and ATP-synthase-dependent respiration, which promotes robust insulin secretion [3]. In addition, mitochondrial Ca^2+^ rise has been suggested to contribute to the improvement of glucose-stimulated insulin secretion by sulfonylurea compounds in type 2 diabetic patients [4]. Finally, genetic evidences have recently demonstrated the high impact of mitochondrial Ca^2+^ uptake for pancreatic β-cell signal transduction and therefore insulin secretion [5,6,7]. Indeed, the molecular nature of the mitochondrial calcium uniporter (MCU), the transporter which mediates the mitochondrial Ca^2+^ influx, was recently disclosed [8,9], opening new perspectives for investigation and molecular intervention [10]. In mouse pancreatic β-cells, consistently with a role of mitochondrial Ca^2+^ on metabolism/secretion coupling, MCU knockdown displayed strong reduction of mitochondrial Ca^2+^ uptake, accompanied by impaired ATP production [7]. Similar results were obtained in pancreatic β-cell lines (INS-1 and INS-1E cells). The genetic depletion of two subunits of the MCU complex (Mcu and Micu1) were characterized by diminished mitochondrial Ca^2+^ transients, impaired respiration rate and ATP levels, and reduced insulin secretion, during glucose stimulation [5,6]. 

While the impacts of Ca^2+^ homeostasis [11,12] and mitochondrial function [13] on β-cell function are widely accepted, and despite the unquestionable role of mitochondrial Ca^2+^ for pancreatic β-cell signal transduction [5,6,7], very few therapies or interventions based on the modulation of mitochondrial Ca^2+^ have been developed until now. In this context, we recently demonstrated that natural bioactive quinic acid increases glucose-stimulated mitochondrial Ca^2+^ rise, enhancing insulin secretion in both INS-1E β-cells and mouse islets, and improves glucose tolerance in mice [14]. In that study, quinic acid was not a direct activator of MCU but it enhanced the release of Ca^2+^ from the endoplasmic reticulum, improving Ca^2+^ transmission between the endoplasmic reticulum and mitochondria. A more specific MCU-targeted pharmacology is expected to improve β-cell MCU activation, promoting beneficial effects in the context of diabetes treatment. Interestingly, some natural plant flavonoids often present in the diet, and especially kaempferol, have been previously reported to direct activate mitochondrial Ca^2+^ uptake in HeLa cells [15]. Moreover, several flavonoids, which are members of the natural bioactive polyphenol family, have been proposed to strengthen survival processes and insulin secretory capacity of β-cells (reviewed in [16]). The idea that flavonoids can preserve the function and survival of β-cells is based on the antioxidant and anti-inflammatory properties of these polyphenols, as well as their ability to stimulate pro-survival pathways and to improve bioenergetics function. However, little attention has been devoted to the possibility of targeting mitochondrial Ca^2+^ uptake with polyphenols to promote metabolism/secretion coupling in pancreatic β-cells.

To determine the impact of a mitochondrial Ca^2+^-targeted intervention on pancreatic β-cell metabolism/secretion coupling, we measured here the effect of the paradigmatic polyphenol putative MCU activator kaempferol on mitochondrial Ca^2+^ rise, insulin secretion, and cell death, during glucose stimulation, in pancreatic INS-1E β-cells. In addition, by taking advantage of the recently developed MCU pharmacology, we quantified the effect of the MCU inhibitor mitoxantrone on mitochondrial Ca^2+^ rise and granule exocytosis in both glucose-stimulated and kaempferol-potentiated pancreatic INS-1E cell function. Finally, we validated the effect of kaempferol on insulin secretion in a model of human pseudo-islets. We proposed to target mitochondrial Ca^2+^ with kaempferol as a nutritional non-cytotoxic intervention, to promote metabolism/secretion coupling in pancreatic β-cells.

## 2. Materials and Methods 

### 2.1. Materials

Chemicals were from Sigma-Aldrich (Buchs, Switzerland), Invitrogen (Life Technologies, Zug, Switzerland), or VWR (Zug, Switzerland), unless otherwise indicated. Kaempferol was dissolved in DMSO at a concentration of 1 mM and then diluted at 10 µM in Krebs-Ringer bicarbonate Hepes buffer (KRBH), containing (in mM): 140 NaCl, 3.6 KCl, 0.5 NaH_2_PO_4_, 0.5 MgSO_4_, 1.5 CaCl_2_, 10 Hepes, and 5 NaHCO_3_; pH 7.4. Glucose was 2.5 mM unless otherwise specified. Drs. T. Pozzan and R. Rizzuto (University of Padova, Padova, Italy) kindly provided the plasmid containing the mitochondrial aequorin mutant to measure mitochondrial [Ca^2+^]. Some figures contain illustrations made by Servier Medical Art, http://www.servier.fr/servier-medical-art.

### 2.2. Cell Culture, Mitochondrial Isolation, and Western Blots

INS-1E cells [17] were cultured at 37 °C in a humidified atmosphere (5% CO_2_) in RPMI-1640 medium (Invitrogen) containing 11 mM glucose, supplemented with 10 mM Hepes (pH 7.3), 10% (v/v) heat-inactivated fetal calf serum (FCS; Brunschwig AG, Basel, Switzerland), 1 mM sodium pyruvate, 50 μM β-mercaptoethanol, 50 μg/mL penicillin, and 100 μg/mL streptomycin (INS-1 medium). Culturing HeLa cells has been previously described [18].

Mitochondria were isolated from INS1E and HeLa cell culture as previously described [19]. Western blots were previously described [20]. The membranes were probed with anti-MCU (Sigma-Aldrich, Buchs, Switzerland), anti-Tom20 (Santa Cruz Biotechnology, Dallas, TX, USA), and anti-ATP-5A (Abcam, Cambridge, United Kingdom) antibodies. 

### 2.3. Mitochondrial Ca^2+^ Measurements

Mitochondrial Ca^2+^ was measured with the genetically encoded luminesce sensor mitochondrial-muted aequorin [15,21]. INS-1E cells were plated on 96-well plates and transfected with the mitochondrial mutant aequorin plasmid, using JetPRIME transfection reagent (Polyplus Transfection, Ilkirch, France). Two days after transfection, cells were washed with KRBH containing 145 mM NaCl, 5 mM KCl, 1 mM MgCl_2,_ 10 mM HEPES, 1 mM CaCl_2,_ and 10 mM glucose; pH 7.4. Then, the cells were incubated with 5 µM coelenterazine native (Biotium, Fremont, CA, USA), dissolved in aequorin buffer, for 2 h. After a second wash with KRBH aequorin buffer, each well was treated for 30 min with DMSO 1% (control) or 10 µM kaempferol or 50 µM mitoxantrone. Luminescence was measured with a Cytation3 plate reader (Biotek, Sursee, Switzerland). The basal luminescence was measured for 30 s, then cells were stimulated for 90 s with glucose (16.7 mM). The experiment was ended by the addition of a digitonin (25µM)/CaCl_2_ (40 mM) solution to calibrate the fluorescence signal, as described by Montero and collaborators [15]. The data per well were analyzed as previously described [22]. The analysis protocol first removed all unused wells than proceeded with a calibration of the data followed by a curve fitting and peak-finding algorithm. Calibration was calculated using the following equation:$$\left[{\mathrm{Ca}}^{2+}\right]\;=\;\frac{{\left(\frac{L}{L_{max}}\;\times \;\lambda \right)}^{\frac{1}{n}}+\;\left({\left(\frac{L}{L_{max}}\;\times \;\lambda \right)}^{\frac{1}{n}}\;\times K_{TR}\right)-1}{K_{R}-{\left(\frac{L}{L_{max}}\;\times \;\lambda \right)}^{\frac{1}{n}}\times \;K_{R}}$$
where L is the luminescence value at sampling time, L*_max_* is the total luminescence emitted up until the sampling time, and *K_R_* and *K_TR_* are the constants for calcium-bound state and the calcium-unbound state, respectively. Finally, λ is the rate constant for aequorin consumption at saturating Ca^2+^ concentration and n is the number of Ca^2+^-binding sites. The calibrated values were then fitted using a spline interpolation, and the area under the curve (AUC) was computed on the fitted curve to serve as the primary read-out for the calcium.

### 2.4. Static Insulin Secretion in INS-1E Cells

INS-1E cells were plated on a polyornithine-treated 24-well plate. After 48 h, cells were washed in Krebs-Ringer bicarbonate Hepes buffer (KRBH), containing (in mM) 140 NaCl, 3.6 KCl, 0.5 NaH_2_PO_4_, 0.5 MgSO_4_, 1.5 CaCl_2_, 10 Hepes, and 5 NaHCO_3_, pH 7.4 containing 2.5 mM glucose. Cells were stimulated for 30 min with 16.7 mM glucose and supernatants were collected. Cellular insulin content was extracted with a mixture of ethanol (75%) and HCl (1.5%) overnight at 4 °C. Insulin was measured using a Rat Insulin Enzyme Immunoassay kit (SpiBio-Bertin Pharma, Montigny le Brotonneux, France).

### 2.5. Cell Death

Kinetic experiments of apoptosis were performed with the IncuCyte ZOOM instrument (Essen Bioscience, Ann Arbor, MI, USA). INS-1E cells were seeded at 50% confluence in 96-well-plate format in RPMI-1640 medium. Then, 24 h later and just before the beginning of the acquisition, the cells were incubated with IncuCyte Annexin-V Green (4642), according to supplier’s instructions. Annexin-V is commonly used to detect apoptotic cells by its ability to bind to phosphatidylserine, an early marker of apoptosis, when it is on the outer leaflet of the plasma membrane (e.g., [23]). Four images per well were collected at the indicated time using a 10× objective and bandwidth filters (Ex: 440/80 nm; Em: 504/44 nm). For image segmentation and processing definition, the following constraints were applied—Parameter: Top-Hat (background subtraction); Radius: 50 µm; Threshold: 0.800 GCU; Edge: 0; Fill: 0; Area: 0; Eccentricity: 0; Mean intensity: 0; and Integrated intensity: 0. Data were exported as area (µm^2^) per well covered by Annexin-V positive objects.

### 2.6. Human Islets and Insulin Secretion

3D InSight™ Human Islet Microtissues were obtained from InSphero AG (Schlieren, Switzerland). Upon arrival, the pseudo-islets were treated following manufacturer’s instructions for measurement of glucose-stimulated insulin secretion. Briefly, the isolated pseudo-islets were carefully washed twice and incubated for 1 h at low glucose in modified Krebs-Ringer buffer (KRB) containing (in mM) 131 NaCl, 4.8 KCl, 1.3 CaCl_2_, 1.2 KH_2_PO_4_, 1.2 MgSO_4_, 5 NaHCO_3_, 25 HEPES, and 2.8 glucose, and 0.5% BSA. This solution is referred to as low glucose solution (LGS, 2.8 mM). Then, isolated islets were carefully washed with LGS and incubated for 2 h in fresh KRB containing 16.7 mM glucose and 0.5% BSA in presence of test substances at indicated concentrations. Measures of insulin levels in supernatants and cell extracts (extraction in acid ethanol (1.5% (v/v) HCl in 70% (v/v) ethanol) were performed using a sensitive chemiluminescence enzyme-linked immunosorbent assay (ALPCO, Salem, NH, USA). All experiments with human islets were approved by the Ethical commission of the Human Research Act (Bern, Switzerland).

### 2.7. Statistics

Data are expressed as means ± SEM. Statistically significant differences between means were determined using Student’s *t*-test for comparison between two means. ANOVA test was used when comparing mean values from multiple groups. Additionally, *p*-values < 0.05 were accepted as significant and represented as * in respective figure panels (NS, not significant, were indicated). 

## 3. Results

### 3.1. MCU Is Expressed in Pancreatic INS-1E β-Cells, Irrespectively of Acute Kaempferol Treatment

To assess the possibility of targeting the mitochondrial Ca^2+^ in pancreatic β-cell metabolism/secretion coupling, the expression level of MCU protein was investigated in a model of glucose-sensitive pancreatic β-cells, named INS-1E β-cells [17]. As shown in Figure 1A, MCU was highly expressed in mitochondria isolated from INS-1E cells. The MCU expression level was also detected in rat mitochondria isolated from brown adipose tissue and brain as control tissues. The impact of the flavonoid kaempferol (Figure 1B) on MCU expression was then quantified in INS1-1E β-cells (Figure 1C). In addition, we checked the effect of the acute treatment with kaempferol on HeLa cells, a system in which mitochondrial calcium uptake is activated by that polyphenol [10]. As shown in Figure 1D, MCU was expressed in both INS-1E and HeLa cells and the acute incubation with kaempferol did not influence the expression level of that mitochondrial transporter. These results validated INS-1E as a good system to study the effect of a mitochondrial Ca^2+^-targeted intervention in a cellular model of glucose-stimulated metabolism/secretion coupling activation. 

### 3.2. Kaempferol Enhances Glucose-Stimulated Mitochondrial Ca^2+^ Rise and Mitoxantrone Inhibits Matrix Ca^2+^ Elevation in Pancreatic Insulin-Secreting Cells

To investigate the effect of a mitochondrial Ca^2+^-targeted intervention on insulin-secreting cells, we measured the effect of kaempferol on mitochondrial Ca^2+^ rise in glucose-stimulated INS-1E cells, transfected with the genetically encoded sensor-mutated aequorin, targeted to mitochondria. As shown in Figure 2A, stimulation with 16.7 mM glucose promoted robust mitochondrial Ca^2+^ elevation in control INS-1E cells, treated with DMSO. Acute treatment with 10µM kaempferol (30 min incubation) potentiated mitochondrial Ca^2+^ rise. Therefore, this treatment increased the integrated mitochondrial Ca^2+^ elevation evoked by glucose stimulation by 37 % (Figure 2B) and the amplitude of mitochondrial Ca^2+^ signal by 27% (Figure 2C). Then, we used mitoxantrone (50 µM), which has been recently identified as a direct inhibitor of MCU [24]. As shown in Figure 2D, mitoxantrone strongly reduced the mitochondrial Ca^2+^ elevation evoked by glucose, the integrated mitochondrial Ca^2+^ response (Figure 2E) and the amplitude of mitochondrial Ca^2+^ signal (Figure 2F) being reduced by 55% and 40%, respectively. These data support the possibility of targeting the mitochondrial Ca^2+^ transport in pancreatic insulin-secreting cells with the natural bioactive kaempferol.

### 3.3. Kaempferol Potentiates Glucose-Stimulated Insulin Secretion

Mitochondrial Ca^2+^ activation has been demonstrated to be an activating signal for insulin secretion [2] via activation of mitochondrial Ca^2+^-dependent processes [3]. Given the positive effect of kaempferol on glucose-stimulated mitochondrial Ca^2+^ rise, we investigated the effect of this flavonoid on insulin secretion. We found that in control cells glucose stimulation promoted robust release of insulin (Figure 3A). 

Kaempferol (10 µM) strongly potentiated glucose-induced hormone exocytosis, the insulin release being increased by 70% (in parallel with the increased percentage of the secreted insulin, inset Figure 3A). To confirm this relationship between mitochondrial calcium transport modulation and insulin secretion, we measured hormone exocytosis in the presence of the MCU inhibitor mitoxantrone. When INS-1E cells were stimulated with 16.7 mM glucose (Figure 3B), no additional insulin release was recorded, indicating that pharmacological inhibition of MCU prevents glucose-stimulated hormone exocytosis. Consistent with genetic manipulation of MCU, these data demonstrate that pharmacological/nutritional intervention on MCU can modulate metabolism/secretion coupling in insulin-secreting cells.

### 3.4. Acute Kaempferol-Dependent Mitochondrial Ca^2+^ Rise Is Not Cytotoxic

Transient increase of mitochondrial calcium uptake promotes cellular energetics [25] and modulates metabolism/secretion coupling [4] in physiological condition. However, a massive increase of mitochondrial calcium uptake regulates cell survival, promoting cell death [10,26]. To investigate if kaempferol-potentiated mitochondrial calcium uptake is associated with cell death in insulin-secreting cells, we measured apoptosis (Figure 4A). 

Acute incubation of kaempferol (30 min) did not enhance phosphatidylserine exposure, an early marker of apoptosis (Figure 4B), indicating absence of cytotoxicity. Interestingly, absence of cytotoxicity was also recorded in cells acutely treated with mitoxantrone, suggesting that the mitoxantrone-induced insulin release is not linked to cell death but it depends on other unspecific effects of this compound. After 24 h, kaempferol (or its metabolites) slightly increased cell death, compared to control (DMSO), whereas mitoxantrone and staurosporine triggered apoptosis, as expected. These effects were amplified after chronic treatment (48 h, Figure 4A,B). These data reveal that acute treatment with kaempferol boosts physiological release of insulin secretion and that it is not cytotoxic in insulin-secreting cells. 

### 3.5. Mitoxantrone Prevents Kaempferol-Dependent Mitochondrial Ca^2+^ Rise and Inhibits the Potentiation of Insulin Secretion

To confirm that an intervention with kaempferol potentiates mitochondrial calcium rise and therefore insulin secretion via MCU, we measured the effects of kaempferol in insulin-secreting cells treated with the MCU inhibitor mitoxantrone. In glucose-stimulated cells, mitoxantrone inhibited kaempferol-potentiated mitochondrial calcium rise (Figure 5A) and insulin secretion (Figure 5B). 

No acute cytotoxic effect was recorded under these experimental conditions (Figure 5C). These experiments indicate that an intervention on MCU aimed to modulate β-cell function is not a simple trademark of kaempferol treatment; instead, this polyphenol enhances metabolism/secretion coupling via MCU activation (Figure 5D). 

### 3.6. Kaempferol Potentiates Insulin Secretion in Hhuman islet Beta Cells

To validate the impact of kaempferol on the potentiation of insulin secretion in a relevant human model of pancreatic β-cells, we measured glucose-induced insulin secretion in standardized human single islet microtissues, derived from primary islet (Figure 6A, 3D InSight™ Human Islet Microtissues). We refer to these tissues as pseudo-islets. Pseudo-islets are uniform in size, cellular composition, and function. In the control experiment, single pseudo-islets were stimulated with glucose 16.7 mM (Figure 6B, left). As expected, glucose stimulation promoted robust release of insulin. Kaempferol (10 µM, Figure 6B, right) also strongly potentiated glucose-induced hormone secretion in pseudo-islets, the insulin release being increased by 57%. These data indicate that the effect of kaempferol in human tissue is consistent with the results obtained in INS-1E cells. 

## 4. Discussion 

Diabetes is a leading cause of morbidity and mortality worldwide. Targeting pancreatic β-cell signal transduction has been proposed as an effective therapeutic approach in both type 1 and type 2 diabetes [1]. Mitochondrial calcium uniporter represents a novel molecular target in this context, due to the ability of matrix Ca^2+^ to modulate metabolism/secretion coupling in the β-cell [2,3]. Genetic and pharmacological evidences [5,6,7] have already reproducibly demonstrated the impact of MCU for β-cell signal transduction. In addition, the mechanism of mitochondrial Ca^2+^-dependent β-cell activation has been previously investigated [2,3]. However, very few interventions based on the modulation of mitochondrial Ca^2+^ have been developed until now. By taking advantage of the recent molecular discovery of the MCU, the mitochondrial channel which mediates matrix calcium transport, and its pharmacology, here we efficiently demonstrated how an intervention on that channel improves glucose-stimulated signal transduction and therefore promotes hormone secretion, in a model of insulin-secreting β-cells (Figure 7). In addition, we validated the effect of the mitochondrial Ca^2+^ activator kaempferol on the potentiation of glucose-induced insulin secretion in a model of human islet beta cells. 

Given the ability of the natural bioactive kaempferol to activate mitochondrial calcium uptake in HeLa cells [15], we investigated the efficacy of that polyphenol also in a model of pancreatic insulin-secreting cells. The binding site of the kaempferol to MCU complex is unknown. The composition and stoichiometry of the MCU components (MCU/MCUb) and of the associated proteins (MICU 1, 2, and 3, EMRE, and MCUR1) is tissue specific and a large variety of mitochondrial Ca^2+^ currents have been observed across different tissues and cell types [27,28]. The relative abundance of distinct MCU subunits and regulators has been demonstrated to affect mitochondrial Ca^2+^ transport in different tissues (e.g., [29]), suggesting a possible tissue-specific effect of kaempferol. Here, we demonstrated that this natural flavonol activates MCU also in insulin-secreting cells. In addition, kaempferol-evoked matrix Ca^2+^ rise was prevented by the pharmacological inhibition of the uniporter (e.g., with mitoxantrone), indicating that the use of kaempferol can be applied to modulate MCU-dependent function in the pancreatic β-cell.

The impact of kaempferol treatment on β-cell function results in a non-cytotoxic potentiation of glucose-stimulated insulin secretion. By targeting the basic molecular machinery of metabolism/secretion coupling, kaempferol promotes hormone secretion via MCU activation. On the other hand, mitochondrial Ca^2+^ overload is well known to also promote cell death pathways [10,26]. Therefore, a massive rise of mitochondrial matrix Ca^2+^ has been established to open the permeability transition pore (review in [30]), a high conductance inner membrane channel [31]. Persistent opening of the permeability transition pore was demonstrated to cause the remodeling of mitochondrial inner membrane cristae, triggering the release of pro-apoptotic proteins (e.g., cytochrome c) and initiation of cell death [32]. We verified the absence of β-cell cytotoxicity by acute treatment with kaempferol, by measuring an early marker of apoptosis (e.g., exposure of phosphatidylserine on the cell surface). The transient increase of kaempferol-dependent mitochondrial Ca^2+^ rise activates mitochondria without promoting cell death. Therefore, matrix Ca^2+^ was extruded immediately after the uptake (Figure 2). Interestingly, prolonged in vitro administration of that natural flavonol (24–48 h) promoted dose-dependent cell death. However, one has to keep in mind that the in vitro reactive polyphenols are antioxidant compounds that can act as or can be converted to prooxidant molecules [33,34], especially in cell culture media [35]. These conditions are not likely to occur in vivo upon consumption of flavon-containing foods, since the bioavailability of polyphenols in foodstuffs is low and metabolism by enterocytes and hepatocytes is very effective [34].

Kaempferol is a plant-derived compound. Plant products are very attractive molecules to target pancreatic β-cells for the treatment of diabetes [36]. These compounds are more attractive than synthetic drugs because of their diversity and minimal side effects. Several natural products and extracts and their bioactive compounds are known to promote beneficial effects on pancreatic beta-cell function [36] and growing amount of evidence supports their efficacy for prevention and attenuation of diabetes consequences [16]. However, in parallel with efficacy and safety evaluations, it is necessary to investigate their mechanism of action as an antidiabetic phytochemical. Kaempferol has previously been proposed to ameliorate streptozotocin-induced diabetes via promoting glucose metabolism in skeletal muscle and inhibiting gluconeogenesis in the liver [37]. In addition, kaempferol treatment has been demonstrated to promote cytoprotective effects in β-cell and human islets exposed to chronic high glucose [38]. These effects were associated with improved insulin secretory function and synthesis. More recently, kaempferol has been proposed to improve hyperglycemia, glucose tolerance, and blood insulin levels and to preserve pancreatic beta-cell mass in obese diabetic mice. Here, we demonstrated a direct effect of this flavonol on the basic molecular machinery of metabolism/secretion coupling of the pancreatic β-cell, via activating glucose-stimulated mitochondrial Ca^2+^ rise. 

To validate an MCU-targeted intervention of kaempferol in pancreatic β-cells, we used mitoxantrone, which was recently identified as a direct inhibitor of MCU [24]. Our results demonstrate that the acute effect of this drug on glucose-stimulated cells and in kaempferol-treated cell cultures strongly supports a causality between mitochondrial Ca^2+^ modulation and pancreatic β-cell activation. Concerning the relevance of the kaempferol concentration used in this investigation, several studies have demonstrated different levels of kaempferol in plasma after treatment. Depending on the source and on the study design, the maximum plasma concentration of kaempferol was recorded in the range of 10/100 nanomolar in some studies (e.g., [39,40,41]). In another study [42], a single dose of 600 mg/kg of standardized kaempferol-containing Ginkgo biloba extract resulted in maximum plasma concentration of kaempferol in the micromolar range, which reflects the concentration range used in our experiments.

The flavonol kaempferol was selected in this study because it has been previously identified to activate mitochondrial Ca^2+^ uptake in HeLa cells [15]. However, in the same study, other natural flavonols were investigated and showed distinct efficacies to potentiate mitochondrial Ca^2+^ uptake. As previously mentioned, mitochondrial Ca^2+^ has been proposed to activate pancreatic β-cells by co-regulating oxidative metabolism (e.g., Nicotinamide adenine dinucleotide, NADH, production) and ATP-synthase-dependent respiration [3]. The oxidative metabolism is driven by the mitochondrial Ca^2+^-dependent activation of matrix dehydrogenases, which are involved in the direct supply of NADH and FADH (Flavin adenine dinucleotide) [43]. Given that the activity of mitochondrial dehydrogenases is sensitive of the mitochondrial Ca^2+^ level [43] and that different polyphenols promote distinct levels of mitochondrial Ca^2+^ uptakes, we speculate that different flavons could promote distinct levels of β-cell activations and therefore insulin secretion and might have similar or even better beneficial effects than kaempferol.

## 5. Conclusions

In summary, we provided evidence for the ability of kaempferol to boost metabolism/secretion coupling in a model of pancreatic β-cells, via potentiation of mitochondrial Ca^2+^ rise (Figure 7). These results, in combination with the non-cytotoxic acute effect of this natural bioactive and the potentiated glucose-induced insulin secretion in human pseudo-islets, indicate that the dietary intake of kaempferol may be developed as an adjuvant to support existing therapies for the treatment of diabetes. Conclusive preclinical and clinical evidences are the next step to validate the efficacy and safety of this phytochemical-based intervention. 

## Figures and Tables

**Figure 1 nutrients-12-00538-f001:**
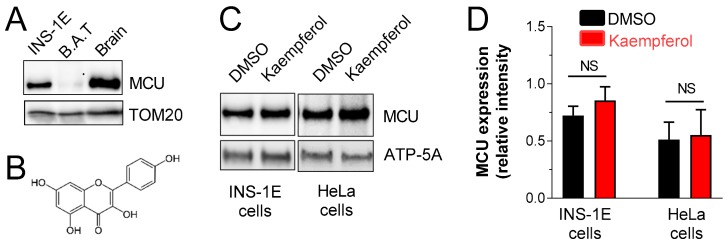
Mitochondrial calcium uniporter (MCU) is expressed in pancreatic INS-1E β-cells and its abundance is not modulated by acute treatment with kaempferol. Detection of MCU by Western blot. (**A**) Mitochondria were isolated from INS-1E cells, rat brown adipose tissue (BAT) or rat brain and 30 μg of proteins were analyzed. TOM20 was detected as control of mitochondrial fraction. (**B**) Structure of kaempferol. (**C**,**D**) Detection (**C**) and quantification (**D**) of the effect of kaempferol on MCU expression, in INS-1E and HeLa cells. Cells were incubated for 30 min with 10 μM kaempferol and then mitochondria were isolated and 30 μg of proteins were analyzed. (**C**) Representative blot from three independent experiments is shown. (**D**) The bars of MCU expression indicate optical density of the respective bands of panel C, which are normalized according the corresponding mitochondrial ATP-5A band. They are the mean ± S.E.M of 3 independent experiments (Student’s *t*-test). NS, not significant.

**Figure 2 nutrients-12-00538-f002:**
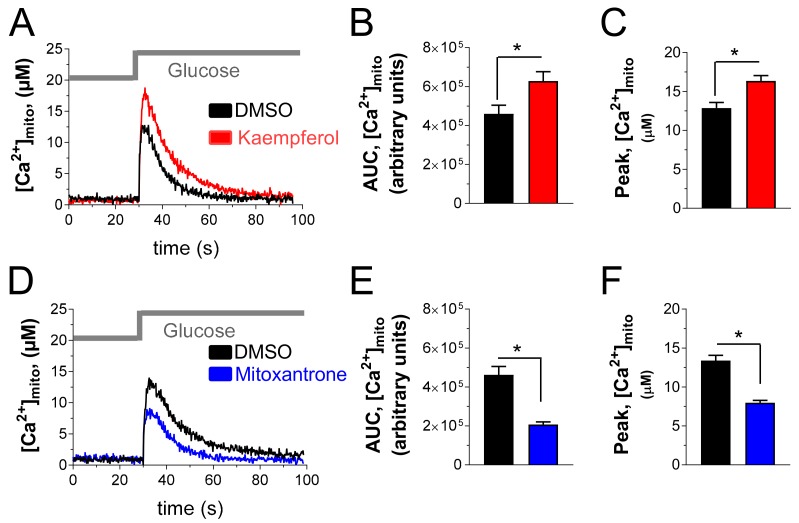
MCU activator kaempferol and MCU inhibitor mitoxantrone increases and inhibits, respectively, glucose-stimulated mitochondrial Ca^2+^ rise in INS-1E cells. (**A**,**C**) Mitochondrial-targeted mutated aequorin was reconstituted with native coelenterazine in INS-1E cells, as described. Then cells were treated for 30 min with 1% DMSO or 10 μM kaempferol (**A**), or 50 μM mitoxantrone (**D**), were placed in the plate reader and stimulated with 16.7 mM glucose, as indicated (Glucose). (**B**,**C**,**E**,**F**) Statistical evaluation of the effect of kaempferol (**B**,**C**) and mitoxantrone (**E**,**F**) on the integrated mitochondrial Ca^2+^ elevation (**B**,**E**) and on the amplitude of the mitochondrial Ca^2+^ signal (C,F), evoked by glucose stimulation. Data are representative (**A**,**D**) or the mean ± SEM (**B**,**C**,**E**,**F**) of 7 independent experiments (Student’s *t*-test. *, *p* < 0.05).

**Figure 3 nutrients-12-00538-f003:**
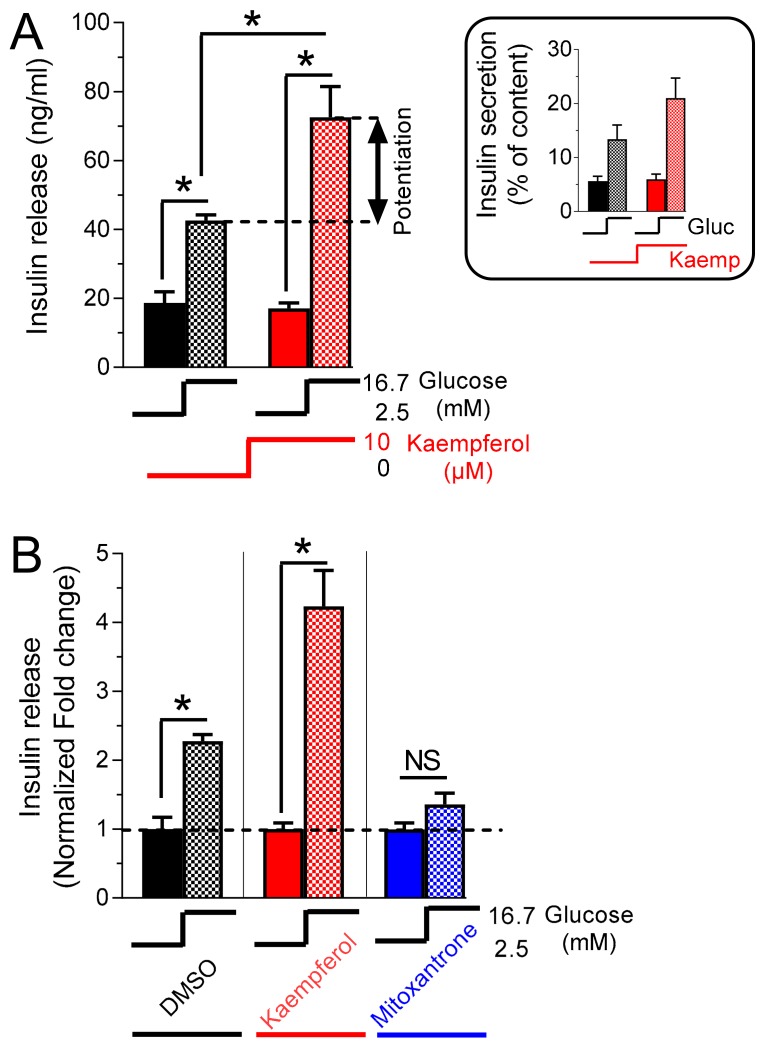
Kaempferol potentiates glucose-stimulated insulin secretion, whereas mitoxantrone inhibits glucose-stimulated hormone exocytosis. Static insulin secretion measurements. (**A**) Insulin release. Control (1% DMSO, black) and kaempferol (10 μM)-treated (red) INS-1E cells were incubated for 30 min in the presence of resting (2.5 mM) and stimulatory (16.7 mM) glucose (Glc) concentrations. Inset, secreted insulin expressed as a percentage of content. Shown is the average of 6 experiments (mean ± SEM; one-way ANOVA test. *, *p* < 0.05). (**B**) Effect of 30 min treatment of mitoxantrone (50 μM, blue) and kaempferol (10 μM, red) on glucose-stimulated (16.7 mM glucose) insulin secretion (insulin release), expressed as a fold change of the respective insulin secretion in resting glucose (2.5 mM), set to 1 for each treatment (DMSO; kaempferol; mitoxantrone). Shown is the average of 6 experiments (mean ± SEM; Student’s *t*-test for the main panel and one-way ANOVA test for the inset. *, *p* < 0.05). NS, not significant.

**Figure 4 nutrients-12-00538-f004:**
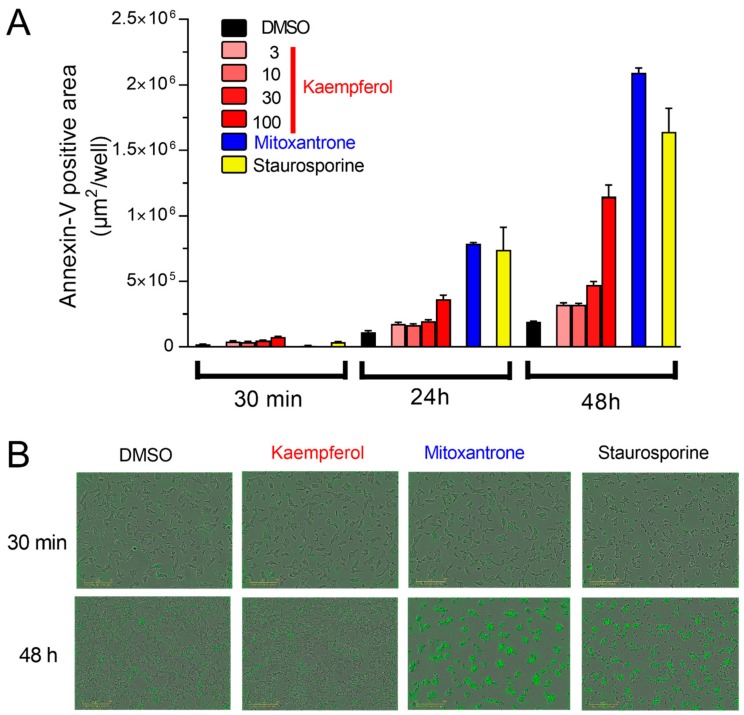
Acute kaempferol-dependent mitochondrial Ca^2+^ rise does not promote cell death. Apoptotic cell death was measured as Annexin-V positive area. (**A**) Quantification of apoptosis measured at 30 min, 24 h, and 48 h in INS-1E cells treated with 1% DMSO (black), kaempferol at the indicated concentrations (in μM, red), 50 μM mitoxantrone (blue), or 100 nM staurosporine (yellow). Representative results are shown (mean ± SEM of 3 replicates). The experiment was repeated 3 times with comparable results. (**B**) Representative images of the data quantified in A, at 30 min and at 48 h; 1%, DMSO, 10 µM kaempferol, 50 µM mitoxantrone and 100 nM staurosporine were indicated.

**Figure 5 nutrients-12-00538-f005:**
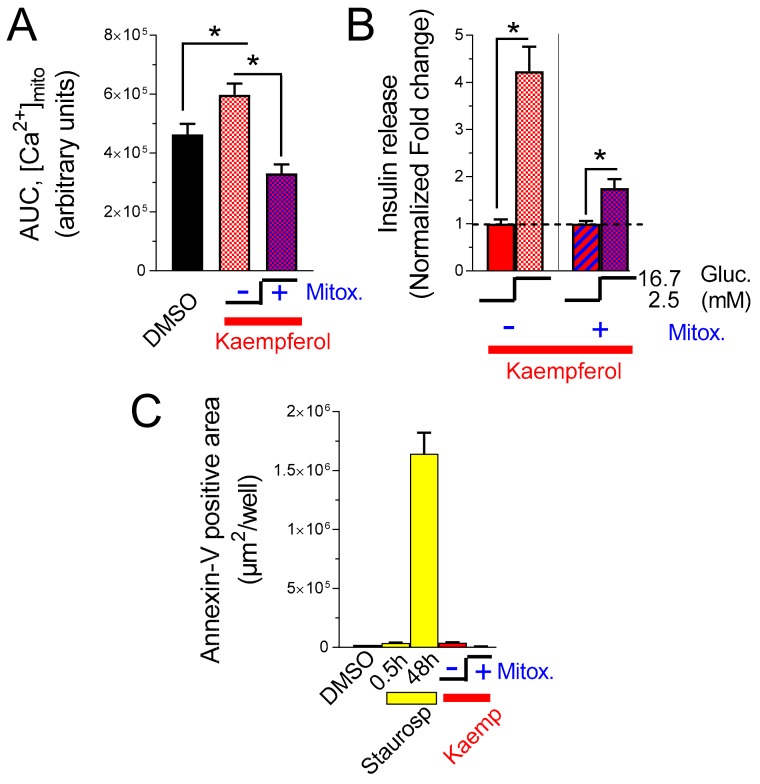
Kaempferol-induced mitochondrial Ca^2+^ rise and insulin secretion are prevented in mitoxantrone-treated cells. (**A**) Mitochondrial-targeted mutated aequorin was reconstituted with native coelenterazine in INS-1E cells. Then, the cells were treated for 30 min with 1% DMSO or 10 µM kaempferol in absence (−) or presence (+) of 50 µM mitoxantrone. Finally, the cells were placed in the plate reader and stimulated with 16.7 mM glucose. The bars represent the statistical evaluation of the integrated mitochondrial Ca^2+^ elevation evoked by glucose stimulation. Data are the mean ± SEM of 10 independent experiments (one-way ANOVA test. *, *p* < 0.05). (**B**) Insulin release was quantified as described in Figure 3B in INS-1E cells treated for 30 min with 1% DMSO or 10 μM kaempferol in absence (−) or presence (+) of 50 µM mitoxantrone. Shown is the average of 6 experiments (mean ± SEM; Student’s *t*-test). (**C**) Apoptosis measured, as described in Figure 4, at 30 min, in INS-1E cells treated with 1% DMSO (black), or 10 µM kaempferol in absence (−) or presence (+) of 50 μM mitoxantrone. Staurosporine 100 nM is shown as a positive control of cell death (yellow) and it is quantified at 0.5 h and 48 h, as indicated. Representative results are shown (mean ± SEM of 3 replicates. *, *p* < 0.05). The experiment was repeated 3 times with comparable results.

**Figure 6 nutrients-12-00538-f006:**
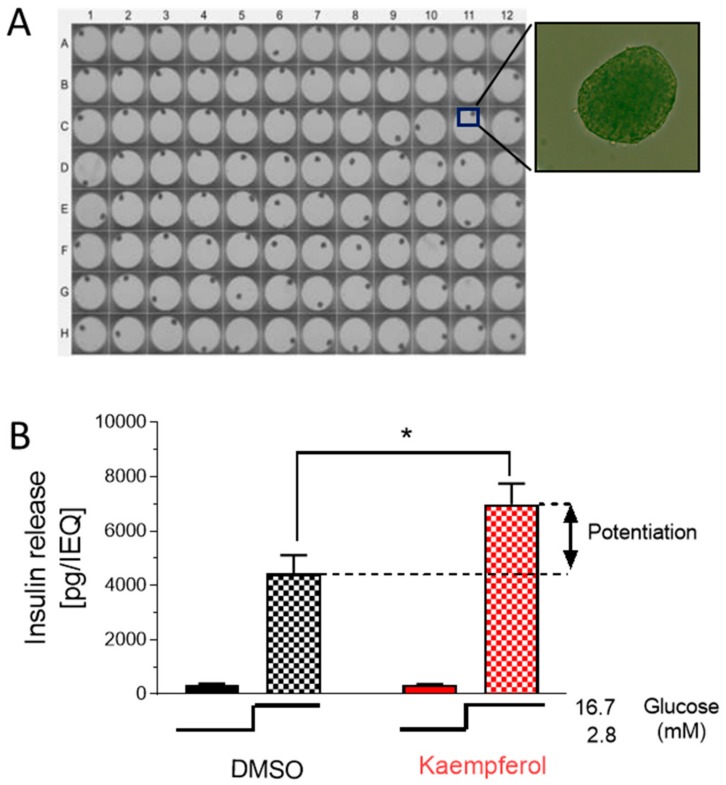
Kaempferol potentiates glucose-stimulated insulin secretion in human pseudo-islets. (**A**) Single isolated human pseudo-islets in 96-well plates. (**B**) Glucose-stimulated insulin secretion of human pseudo-islets exposed to low glucose (2.8 mM, *n* = 6) or high glucose solution (16.7 mM, *n* = 12) in combination with either DMSO 0.1% (black) or 10 μM kaempferol (red). Shown is the mean ± SEM. Statistical analysis was performed by ANOVA test (*, *p* < 0.05).

**Figure 7 nutrients-12-00538-f007:**
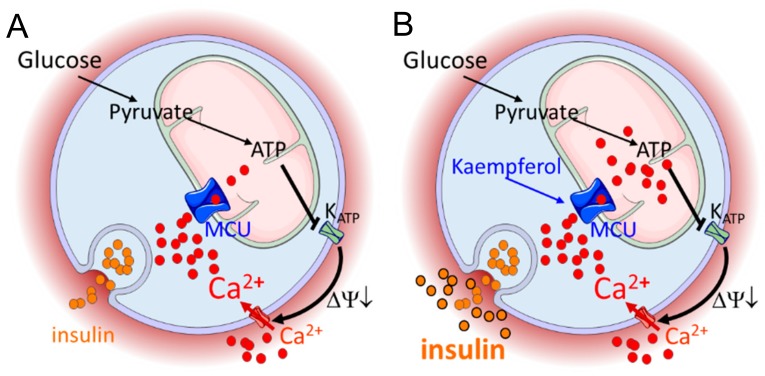
Proposed mechanism of kaempferol-dependent metabolism/secretion coupling in glucose-stimulated pancreatic β-cells. The glucose-stimulated insulin secretion (**A**) is potentiated by kaempferol, via mitochondrial calcium rise enhancement (**B**).

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
