# Peer review of "Targeting Mitochondrial Calcium Uptake with the Natural Flavonol Kaempferol, to Promote Metabolism/Secretion Coupling in Pancreatic β-cells"

_nutrients, 2020, doi:10.3390/nu12020538_

Round 1

Reviewer 1 Report

concerns addressed

Reviewer 2 Report

This referee considers that the manuscript have been improved with the new experiments performed in human pseudo-islets and the text added in the different sections along the manuscript.

This manuscript is a resubmission of an earlier submission. The following is a list of the peer review reports and author responses from that submission.

Round 1

Reviewer 1 Report

In the manuscript titled: “Targeting mitochondrial calcium uptake with the natural flavonol kaempferol, to promote metabolism/secretion coupling in pancreatic insulin secreting cells” Bremont and colleagues present results of the experiments performed on model cell culture cells and demonstrate that flavonoid treatment stimulates insulin secretion though activation of the MCU. This study addresses an important subject related to the mechanisms of flavonoid actions. Taking into account the antioxidant effects of these natural products do not necessarily explain their activities in vivo demonstration of the novel pathway that involves stimulation of mitochondrial calcium uptake is an important step.  Overall, experimental design is straightforward and results support main conclusion. I have several suggestions:

In discussion it will be beneficial in authors would compare concentrations of the compound that they used with concentrations expected to be reached in vivo experiments. In addition to the AUC it will be helpful to have bar graphs with the stats for peak Ca. Do authors expect that other natural flavonoids of the same family might have similar beneficial effects? This can be briefly discussed.

Author Response

Reviewer 1

In discussion it will be beneficial if authors would compare concentrations of the compound that they used with concentrations expected to be reached in vivo experiments.

We have compared the concentrations used in this paper with in vivo data of kaempferol in plasma.Therefore, several studies demonstrated different levels of kaempferol in the plasma after treatment. Depending on the source and on the study design, the maximum plasma concentration of kaempferol was recorded in the range of 10/100 nanomolar in some studies (Calderon-Montano et al., 2011; Radtke et al., 2002; DuPont et al., 2004, for instance). In another study, (Rangel-Ordonez et al., 2010) a single dose of 600 mg/Kg of standardized kaempferol-containing Ginkgo biloba extract, resulted in maximum plasma concentration of kaempferol in the micromolar range, which reflects the concentration range used in our experiments.

In addition to the AUC it will be helpful to have bar graphs with the stats for peak Ca.

We have added the bar graphs with the stats for the Ca2+ peak (Fig 2C and 2F).

Do authors expect that other natural flavonoids of the same family might have similar beneficial effects? This can be briefly discussed.

We have discussed the possibility of an intervention with other natural flavonoids and we have speculated about their possible distinct effects on metabolism/secretion coupling, in function of the differently potentiated mitochondrial Ca2+ elevations.

Reviewer 2 Report

Why the study has been performed in a tumoral insulin secreting cell line instead of mouse pancreatic beta cells? Using mouse or even human pancreatic beta cell, it would have raised its interest, novelty and aplicability for the treatment of diabetes.

Author Response

Why the study has been performed in a tumoral insulin secreting cell line instead of mouse pancreatic beta cells? Using mouse or even human pancreatic beta cell, it would have raised its interest, novelty and aplicability for the treatment of diabetes.

We agree with the referee on the major impact of our results on mouse or human pancreatic beta cells instead of INS-1E cell line, especially in the context of diabetes treatment. However, pre-clinical/clinical impact of kaempferol for the treatment of diabetes are beyond the scope of this paper. The aim of this paper is to determine if an intervention on mitochondrial calcium uptake with a specific polyphenolic MCU activator promotes metabolism/secretion coupling in pancreatic beta cell. To demonstrate this hypothesis we needed a model of glucose-responding pancreatic beta cells, in which we could easily control the environmental condition. In addition, it was necessary to focus exclusively on the beta cell. INS-1E beta cells are a good cellular model of glucose-induced insulin secreting cells. By using this cell line we can bypass the complications linked to the presence of the other beta-cell types (present in mouse or human islet) and clearly demonstrate the relevance of our strategy on a beta cell model. We have now demonstrated the efficacy of a mitochondrial Ca2+-targeted intervention to potentiate glucose-stimulated insulin secretion. This novel evidence is, in our opinion, the prerequisite necessary to design a following study, including an intervention on mouse (or human) islets, as suggested by the referee.

Round 2

Reviewer 2 Report

This referee know and assume that preclinical and clinical impact, for diabetes treatment, of kaempferol is beyond the scope of the manuscript. However, I think that as a model of glucose-responding pancreatic beta cells, the authors should use the best one, which are native pancreatic beta cells within pancreatic islets (mouse or rat). The authors do not need to use a tumoral cell line to control environmental conditions. Pancreatic mouse and rat pancreatic islets have been used during the las four decades as the best model to study the glucose-stimulus secretion coupling and their physiology. In fact, the presence of other non-beta cells within the islet is not a complication, because in native pancreatic islets, beta cells are integrated with other endocrine cell types. In this regard, it is well known, since the early eighties, that beta cell glucose-stimulus secretion coupling cannot be explored and studied without considering the other endocrine cells present in the pancreatic islets. There are hundreds of studies in the literature indicating that pancreatic beta cells work together with the rest of endocrine cell present in the islet. The whole pancreatic islet work as a synthitium. Thus, the clear suggestion of this referee is to perform the study using mouse or rat native and isolated pancreatic islets.

Author Response

We have measured the effect of kaempferol in a model of standardized pancreatic human islet (3D InSigh human islet microtissues). In the new figure 6 we have confirmed that kaempferol potentiates glucose-stimulated insulin secretion also in this human non-tumoral pancreatic tissue.